# The Protective Effect of (-)-Tetrahydroalstonine against OGD/R-Induced Neuronal Injury via Autophagy Regulation

**DOI:** 10.3390/molecules28052370

**Published:** 2023-03-04

**Authors:** Yumei Liao, Jun-Ya Wang, Yan Pan, Xueyi Zou, Chaoqun Wang, Yinghui Peng, Yun-Lin Ao, Mei Fong Lam, Xiaoshen Zhang, Xiao-Qi Zhang, Lei Shi, Shiqing Zhang

**Affiliations:** 1Department of Cardiovascular Surgery, The First Affiliated Hospital, Jinan University, Guangzhou 510632, China; 2JNU-HKUST Joint Laboratory for Neuroscience and Innovative Drug Research, College of Pharmacy, Jinan University, Guangzhou 510632, China; 3Guangdong Provincial Engineering Research Center for Modernization of TCM, College of Pharmacy, Jinan University, Guangzhou 510632, China; 4Centro Hospitalar Conde de São Januário, Macau, China; 5NMPA Key Laboratory for Quality Evaluation of TCM, Jinan University, Guangzhou 510632, China

**Keywords:** (-)-Tetrahydroalstonine, cerebral ischemia, OGD/R, autophagy, Akt/mTOR

## Abstract

Here, (-)-Tetrahydroalstonine (THA) was isolated from *Alstonia scholaris* and investigated for its neuroprotective effect towards oxygen–glucose deprivation/re-oxygenation (OGD/R)-induced neuronal damage. In this study, primary cortical neurons were pre-treated with THA and then subjected to OGD/R induction. The cell viability was tested by the MTT assay, and the states of the autophagy–lysosomal pathway and Akt/mTOR pathway were monitored by Western blot analysis. The findings suggested that THA administration increased the cell viability of OGD/R-induced cortical neurons. Autophagic activity and lysosomal dysfunction were found at the early stage of OGD/R, which were significantly ameliorated by THA treatment. Meanwhile, the protective effect of THA was significantly reversed by the lysosome inhibitor. Additionally, THA significantly activated the Akt/mTOR pathway, which was suppressed after OGD/R induction. In summary, THA exhibited promising protective effects against OGD/R-induced neuronal injury by autophagy regulation through the Akt/mTOR pathway.

## 1. Introduction

Ischemic stroke is responsible for over 80% of all strokes, which are one of the leading causes of mortality and disability worldwide, causing a huge global health and economic burden [1,2]. The approved pharmacological agents for clinic ischemic stroke treatment are limited to thrombolytic agents. While other therapies, including immunomodulatory and cytoprotective agents, have shown promising efficacy in preclinical modeling, none have translated well to the clinic. Complex pathological mechanisms have been reported in neural injury after ischemic stroke, including autophagy dysfunction, oxidative stress, an immune microenvironment, inflammation, etc. Among these, autophagy dysregulation plays a crucial role in neural damage during ischemic stroke [3]. Autophagy is a biological process that degrades and removes cellular materials via the lysosomal pathway [4], while the pathological damage in ischemic stroke is partly caused by autophagy dysfunction via the mTOR-dependent pathway [5]. It was demonstrated that the suitable activation of autophagy is beneficial for neuronal survival, while excessive activation induces neuronal damage under ischemic damage, indicating a two-way effect of autophagy on ischemic stroke [6]. Therefore, the discovery of compounds to modulate autophagy may be a promising therapeutic strategy in ischemic stroke treatment [7].

Monoterpenoid indole alkaloids (MIAs) constitute the most diverse classes in plant alkaloids, which are characterized by structural complexity and intriguing biology [8]. MIAs are widely distributed in the *Apocynaceae* plants, with strong activity of central nervous system excitation and cerebral vasodilation. Our previous study reported that MIAs exhibited vasorelaxant and acetylcholinesterase (AChE) inhibitory activities [9]. In addition, MIAs isolated from *Ervatamia officinalis*, an *Apocynaceae* plant, exhibited promising effects against neuron toxicity induced by oxygen–glucose deprivation/re-oxygenation (OGD/R) [10]. On the other hand, it was reported that MIAs regulated lysosomal and autophagy function via the mTOR pathway [11,12], suggesting that MIAs exhibit promising effects on autophagy regulation. The above studies strongly indicated that identifying lead compounds from MIAs for autophagy regulation may be beneficial for ischemic stroke treatment.

As an evergreen plant of *Apocynaceae*, *Alstonia scholaris* (L.) R. Br. is widely distributed in the tropical regions of Africa and Asia. The leaf of *Alstonia scholaris* has been used as a natural medicine for the treatment of several types of disorders for centuries [13]. Pharmacological studies indicated that *Alstonia scholaris* exhibited a range of biological activities. Specifically, it was reported that the ethanolic extracts of *Alstonia Scholaris* possessed neuroleptic activity in psychotic animal models [14], and its methanol extract exhibited promising neuropharmacological activity in a rat model of neuropathic pain [15]. However, the pharmacological activity and potential molecular mechanism of MIAs from *Alstonia scholaris* on ischemic stroke are far from clear.

In this study, we extracted (-)-Tetrahydroalstonine (THA) from *Alstonia scholaris* with high purity and demonstrated the neuroprotective effect of THA against OGD/R-induced neuronal damage in cortical neurons. Furthermore, we tested the time sequence of autophagy signaling transduction in OGD/R and revealed that autophagic and lysosomal dysfunction were exhibited at the early re-oxygenation phase of OGD/R. THA treatment effectively inhibited early overactivated autophagy signaling transduction and then reduced neuronal death. Overall, our research showed that THA protects against OGD/R-induced ischemic neuronal injury via autophagy regulation, which might be a promising neuroprotective agent used for ischemic stroke treatment.

## 2. Results

### 2.1. THA Alleviates OGD/R-Induced Neuronal Injury

In this study, THA was isolated from *Alstonia scholaris*, with purity of 98.21% (Figure 1A,B). The structural features of THA were further confirmed by NMR spectroscopy (shown in Appendix A, Appendix A). The cytotoxicity of THA was firstly monitored by the MTT assay after different concentrations of THA (12, 6 and 3 µM) were added to cortical neurons for 24 h. The results showed that THA significantly exhibited toxicity towards cortical neurons at the dosage of 12 µM (Figure 1C). Furthermore, to monitor the protective role of THA on OGD/R-induced cortical neuronal injury, THA (3, 1.5 and 0.75 µM) was pretreated for 2 h and cell viability was further monitored after OGD/R induction. Edaravone, a free radical scavenger and a neurovascular protective agent, was used as the positive control according to the reference [16]. As expected, OGD/R induction dramatically reduced the cell viability to around 60% of the control group in primary cortical neurons. THA treatment significantly reversed the decline in a dose-dependent manner, and exhibited a superior therapeutic effect to edaravone (Figure 1D). The results indicated that THA substantially protected neurons against OGD/R-mediated cytotoxicity.

### 2.2. OGD/R Induces Autophagic and Lysosomal Dysfunction at Different Stages

To monitor the changes in autophagic function in cortical neurons during ischemic injury, the autophagic flux markers LC3B and P62’s expression was evaluated by Western blot. The expression of P62 was reported as a marker of autophagic flux impairment in OGD/R induction [17]. As shown in Figure 2A–C, the ratio of LC3B-II/LC3B-I was rapidly up-regulated and P62 was remarkably down-regulated after 1 h of reperfusion, which returned to near basal levels afterward. Moreover, the expression of LAMP1 (lysosomal-associated membrane protein 1), cysteine proteinases CSTB (cathepsin B), and TFEB (a master regulator for lysosome genesis) were assessed to monitor the lysosomal function. The result showed that LAMP1 was increasingly accumulated in cortical neurons with the increase in reperfusion time, and a significant difference was found after 6 h (Figure 2D,E). Meanwhile, the expression of CSTB and TFEB proteins was remarkably reduced after 1 h of reperfusion and gradually returned to near basal levels after 24 h, indicating the occurrence of lysosome-associated dysfunction (Figure 2F,G). In general, these results suggested that autophagic and lysosomal dysfunction were exhibited at the early re-oxygenation phase of OGD/R in cortical neurons.

### 2.3. THA against OGD/R-Induced Neuronal Damage via Promoting Lysosomal Function

To monitor the effect of THA on autophagy regulation in OGD/R-induced neuronal injury, the primary cultured cortical neurons were subjected to OGD and reoxygenation for 1 h with or without THA. As shown in Figure 3A–E, compared with the control group, OGD/R significantly increased LC3B-II/LC3B-I and LAMP1 but decreased CSTB and TFEB protein levels. However, THA administration significantly ameliorated the abnormal expression of these proteins, which might be explained by the THA-mediated activation of lysosomal degradation. The expression of LC3B was also monitored by immunofluorescence. Similarly, the expression of LC3B was up-regulated in the OGD/R group compared to the control group, which was suppressed by THA administration (Figure 3F). To further investigate the role of lysosomal degradation in the THA-mediated therapeutic effect, chloroquine (CQ), a lysosome inhibitor, was used to treat OGD/R-induced neurons with or without THA. As expected, CQ (10 µM) treatment increased the ratio of LC3B-II/LC3B-I in cortical neurons without OGD/R induction. THA administration significantly decreased the ratio of LC3B-II/LC3B-I after OGD/R induction, which was counteracted by CQ (Figure 3G,H). Meanwhile, CQ significantly blocked the THA-mediated increase in cell viability after OGD/R (Figure 3I). Thus, CQ abolished the protective effects of THA, indicating that autophagic lysosomal degradation is required for THA-mediated neuroprotection in OGD/R-induced neuronal injury.

### 2.4. THA Ameliorates the OGD/R-Induced Inhibition of Akt/mTOR Pathway

The activation of the Akt/mTOR pathway was reported to play important roles in both autophagy regulation and ischemic stroke progression [18]. mTOR is an atypical serine/threonine kinase that forms two multiprotein complexes, mTORC1 and mTORC2. mTORC1 is a downstream molecule of AKT and plays a negative regulator role in autophagy [19]. Herein, we monitored the states of the AKT/mTORC1 pathway after THA administration in OGD/R-induced neurons by testing AKT and mTOR activity. As shown in Figure 4, the ratios of p-Akt/Akt and p-mTOR/mTOR were significantly decreased in cortical neurons after OGD/R induction, which was remarkably suppressed by THA administration. THA ameliorated the inactivation of the Akt/mTORC1 pathway induced by OGD/R. The above findings indicated that the therapeutic effect of THA on OGD/R-mediated neuronal damage might be partly accounted for by the Akt/mTORC1 pathway.

## 3. Discussion

Ischemic stroke is a major cause of mortality and disability in the world. Although major advances have recently been made in pharmacological research, unfortunately, tissue plasminogen activator (tPA) is the only FDA-approved drug for the clinical treatment of ischemic stroke to date, and its clinical use is limited by the narrow time window and the risk for hemorrhage. Specific and effective novel drugs are still an urgent need for the treatment of ischemic stroke. Natural compounds, including MIAs, are considered as important sources of drug development for neurological disorder treatment. Previous pharmacological studies demonstrated that MIAs have various biological activities, including anti-tumor [20], anti-viral [21], anti-inflammatory [22], anti-hypertensive [23] and neuroprotective activities. It was also reported that MIAs were found to improve the metabolism of ischemic tissue and protect against neuronal damage [24,25], suggesting that MIAs are promising candidate agents for the treatment of ischemic stroke. 

*Alstonia scholaris* is a well-known source of alkaloids and widely used as a traditional natural medicine. It was reported that the total alkaloids extracted from *Alstonia scholaris* improved nonalcoholic fatty liver disease in a mouse model [26], the total alkaloids also showed the ability to treat respiratory infections in mice with H1N1 and beta-hemolytic streptococcus infectious [13], and the total alkaloids exhibited an inhibitory effect against airway inflammation in rats induced by lipopolysaccharide [27]. These studies suggested that alkaloids from *Alstonia scholaris* have promising biological activities in a wide range of diseases; however, the neuroprotective properties and potential mechanism of alkaloids from *Alstonia scholaris* are far from clear. In the present study, we demonstrated that the THA, a type of MIA isolated from *Alstonia scholaris*, significantly protected cortical neurons against OGD/R-mediated cytotoxicity within the dosage of 0.75–3.0 μM, and the therapeutic effect of THA was superior to that of edaravone. It should be noted that THA was administered before the OGD/R in this study, while drugs are usually taken after stroke in clinical practice. The reason is that cellular damage occurs immediately during the process of OGD/R, but it usually takes time for the drugs to exert an effect via regulating cellular signaling pathways. Therefore, pre-treatment provides a better paradigm to reveal the effects of THA and is widely used for in vitro studies of anti-stroke drugs. The therapeutic effect of THA will be validated via post-modeling administration in animal models in future studies. 

In ischemic stroke, autophagic flux is impaired and leads to the accumulation of autophagosomes, which is detrimental to neuronal survival during reperfusion. Autophagy activation plays a dual role during cerebral ischemic injury. Moderate autophagy promotes neuroprotection and reduces brain damage caused by ischemia, whereas chronic excessive autophagy can lead to neuron death and tissue damage [28,29]. Given that lysosomal function is critical for neuronal survival and might be a promising target to treat ischemia stroke, several types of synthetic and natural compounds were found to protect neurons against ischemic damage through the regulation of the autophagy–lysosomal pathway [30]. In the present study, to determine the dynamic effects of OGD/R on autophagy function, we monitored the expression levels of various autophagy and lysosomal markers (such as LC3B-I/II, P62, TFEB, CTSB) in cortical neurons at different stages after OGD/R induction. LC3B-II, a standard marker for autophagosomes, reflects autophagic activity. The SQSTM1/P62 protein is a ubiquitin-binding scaffold protein and a specific marker of autophagic flux, and soluble P62 was found to be decreased in OGD/R [17]. TFEB is a member of the MiT/TFE3 protein family involved in the regulation of autophagy–lysosomal biogenesis and function, and it was reported that TFEB expression was decreased as a reflection of lysosomal dysfunction during severe ischemic injury [31]. CTSB is a lysosomal protease responsible for degrading damaged organelles or misfolded proteins transported by autophagy, and the expression of CTSB is usually decreased when the cell is exposed to stressful conditions or toxic substances [32]. LAMP1 is abundantly expressed on the lysosomal membrane and used as a marker for lysosomal amounts. In the present study, we found that the LC3B-II/LC3B-I proportion was rapidly elevated and P62 dramatically dropped within 1 h, which were gradually returned to near-normal levels at the late stage, suggesting the activation of autophagy at the early stage after OGD/R. Meanwhile, the expression of TFEB and CSTB was down-regulated at the early stage of reoxygenation and returned to near-normal levels at 6 h after OGD/R in cortical neurons, indicating that OGD/R may induce lysosomal storage dysfunction at the early stage of OGD/R. Meanwhile, the expression of LAMP1 was increased gradually and a significant difference was found at 6 h after OGD/R. Considering that autophagy is a complex and dynamic process, in the present study, we mainly monitored the effect of THA on the autophagic dysfunction of cortical neurons at 1 h after OGD/R, when autophagic dysfunction is maximum. As expected, THA significantly ameliorated the abnormal expression of autophagy and lysosomal markers in cortical neurons at 1 h after OGD/R, suggesting that THA exerts neuroprotective effects via autophagy regulation. Our findings are in accordance with a previous report that the TFEB level was down-regulated by OGD with 1 h reperfusion, and the up-regulation of TFEB by drugs protected neural cells against OGD/R-induced toxicity [7].

Cellular autophagy is known to be regulated by multiple signaling pathways, among which the Akt/mTOR pathway is crucial in autophagy regulation during ischemic stroke. Akt phosphorylation activated mTOR and ultimately inhibited autophagy, and mTOR regulated the degradation capacity of autolysosomes [33]. Specifically, this pathway offers promising insights for the development of new avenues of neurodegenerative disease treatment. mTOR is an atypical serine/threonine kinase, which is normally assembled in two distinct conserved complexes, mTORC1 and mTORC2. mTORC1 is a downstream module of AKT and plays a negative regulatory role in autophagy, while mTORC2 may also regulate autophagy through multiple downstream effector proteins and crosstalk mechanisms, such as the phosphorylation of AGC kinases [34]. In this study, we further demonstrated that THA treatment significantly ameliorated the inactivation of Akt and mTORC1 in cortical neurons after OGD/R, suggesting that the therapeutic effect of THA might be accounted for by the Akt/mTOR pathway. Our findings were consistent with the previous reports that several types of diarylheptanoids, isolated from *Alpinia officinarum*, could attenuate autophagy and demonstrate a neuroprotective effect on cortical neurons against OGD/R via the activation of the AKT/mTOR pathway [35,36]; Panax notoginseng saponins promoted the phosphorylation of AKT and mTOR to alleviate neuronal injury induced by OGD/R [37]. More evidence is still needed to address the specific target of THA in the treatment of ischemic stroke.

In conclusion, the present study demonstrates that THA, a type of MIA extracted from the roots of *Alstonia scholaris*, significantly reduced the neuronal injury induced by OGD/R in cortical neurons. The underlying mechanism might be accounted for by the THA-mediated autophagy regulation through the Akt/mTOR pathway (Figure 5). However, there are some limitations in the present study. Firstly, the toxicity of THA needs to be further assessed to be used in animal models. It was reported that the total alkaloids from the leaves of *Alstonia scholaris* had a wide safety range in both rats and dogs after oral administration in chronic toxicity studies [13], suggesting that the alkaloids from *Alstonia scholaris*, including MIAs, have potential safety as novel drug candidates. Secondly, the blood–brain barrier permeability of THA and its pharmaceutical effect on animal models are still unclear. To address these concerns, the therapeutic effect and safety of THA in an animal model of ischemic stroke will be studied in the future. Meanwhile, autophagy dysfunction has been considered as a common pathological mechanism in many neurological disorders, including stroke, Alzheimer’s disease, Parkinson’s disease, amyotrophic lateral sclerosis, etc. THA, as an autophagy regulator from natural plants, might be a potential protective agent for both brain injury and neurological diseases. 

## 4. Materials and Methods

### 4.1. Isolation and Purification of THA 

THA was isolated from the roots of *Alstonia scholaris* according to the previously described method [38]. The air-dried root power (19.8 kg) of *Alstonia scholaris* was percolated 3 times by 95% EtOH at room temperature. The alcoholic extract was evaporated by pressure reduction to obtain a crude residue (498.7 g), and then the water suspension of the residue was adjusted to pH 2 by using 5% hydrochloric acid. After extraction with CHCl_3_, the acidic water-soluble materials were basified with ammonium hydroxide solution to pH 10 and partitioned with CHCl_3_ to obtain a total alkaloid residue (48.5 g), which was subjected to a silica gel chromatography (200–300 mesh; CHCl_3_/CH_3_OH, 100/0→0/100) to afford 12 fractions (A–L). Fr. B (8.6 g) was purified with Sephadex LH- 20 (CHCl_3_-CH_3_OH, 1:1) to yield 8 subfractions (Fr. B1–Fr. B8). Fr. B2–Fr. B7 were combined and chromatographed on an ODS column with CH_3_OH-H_2_O (10:90 to 100:0, *v*/*v*) to yield 11 subfractions (Fr. B2a–Fr. B2k). Fr. B2d was purified by preparative HPLC (CH_3_OH-[H_2_O + 0.1‰ Et_3_N], 70:30) to afford THA (27.1 mg, *t*R = 20.1 min). Preparative HPLC (pHPLC) was completed on the Agilent 1260 system (equipped with a G1310B Iso pump and G1365D MWD VL detector, Agilent Technologies Inc., Santa Clara, CA, USA) with a Waters Xbridge™ 5 μm C_18_ OBD reversed-phase column (19 × 250 mm, Waters, Milford, MA, USA). The THA was analyzed with a Phenomenex Kinetxe™ 5 μm C_18_ reversed-phase column (4.6 × 250 mm, Phenomenex Inc., Torrance, CA, USA) on the Agilent 1260 system (equipped with a G1311C 1260 Quat Pump VL and G4212B 1260 DAD detector, Agilent Technologies Inc., Santa Clara, CA, USA) to monitor the purity (CH_3_CN-[H_2_O + 0.1‰ Et_3_N], 60:40, tR = 7.66 min).

### 4.2. Chemicals, Reagents and Antibodies

The MTT, poly-l-lysine and chloroquine (CQ) were purchased from Sigma-Aldrich (St. Louis, MO, USA). Anti-LC3B (83506S), anti-SQSTM1/P62 (5114T), anti-ATG5 (12994S), Anti-CSTB (31718), anti-p-Akt (2965S), anti-Akt (4691S), anti-p-mTOR (2971S) and anti-mTOR (2972S) were obtained from Cell Signaling Technology. Anti-TFEB (13372-1-AP) was obtained from Proteintech (Rosemount, IL, USA). Anti-LAMP1 (ab24170) was purchased from Abcam (Cambridge, MA, USA). Neurobasal culture medium, fetal bovine serum (FBS), DMEM culture medium, B27, l-glutamine and d-glucose were purchased from Gibco.

### 4.3. Primary Cortical Neuron Culturing

The cortical neurons were primary cultured according to a previous report [39]. The cortex was collected from Sprague-Dawley rat embryos at day 18 of gestation and digested with trypsin. Primary cortical neurons were cultured with neurobasal medium with 1 mM l-glutamine, 1 mM d-glucose, 2% B27 and 1% penicillin/streptomycin in poly-l-lysine-coated plates. The neurons were maintained for 7 days with half-medium changed every 3 days, and then the following experiments were performed.

### 4.4. OGD/R and Drug Exposure

THA, edaravone and chloroquine were administrated to neurons for 2 h before and 24 h after OGD/R induction. The neurons were cultured with glucose-free DMEM medium in a Modular Incubator Chamber (Billups-Rothenberg, Del Mar, CA, USA) for 4 h with hypoxia environmental conditions (95% N_2_ and 5% CO_2_). Then, cortical neurons were refreshed with the original medium and cultured in the normoxia condition for reperfusion.

### 4.5. Cell Viability Assay

The cortical neurons were incubated with MTT solution in the dark for 3 h and insoluble dark-blue formazan was produced. After being dissolved with DMSO, the optical density (OD) was measured by a multimode detector (Beckman Coulter, Miami, FL, USA) at 595 nm. The cell viability was assessed according to the following formula: cell viability (%) = OD^Sample^/OD^Control^ × 100%.

### 4.6. Western Blot Analysis

The cortical neurons were lysed in RIPA buffer supplemented with protease inhibitor and the supernatant was collected after centrifugation. The Pierce BSA Protein Assay kit (Thermo Scientific, Waltham, MA, USA) was used to determine the protein quantitation. The proteins were separated by 8–12% SDS-PAGE gel and transferred to PVDF membranes. Blocked membranes were incubated with antibodies against LC3B (1:500), SQSTM1/P62 (1:1000), CSTB (1:1000), LAMP1 (1:1000), TFEB (1:1000), p-Akt (1:1000), Akt (1:1000), p-mTOR (1:1000), mTOR (1:1000) overnight at 4 °C. The membranes were then incubated with secondary antibodies. By using with Plus-ECL (enhanced chemiluminescence), the protein bands were detected by the Amersham Imager 600 instrument (Cytiva, Washington, DC, USA) and the quantification was performed by Image J software.

### 4.7. Immunofluorescence

The cortical neurons were fixed in 4% paraformaldehyde and permeabilized with 0.4% Triton X-100. Then, neurons were stained with primary antibodies LC3B (1:150) and MAP2 (1:500) overnight at 4 °C, followed by incubation with Alexa Fluor 488 goat anti-mouse (1:300) and Alexa Fluor 546 goat anti-rabbit (1:500) secondary antibodies. After being stained with DAPI (1:1000) for 10 min and mounted with Hydromount, the immunofluorescence images were detected with a confocal microscope (Zeiss LSM 800, Carl Zeiss Meditec AG, Jena, Germany).

### 4.8. Statistical Analysis

The data were expressed as mean ± standard error of the mean (SEM) from at least three independent experiments. Statistical significance was considered as *p* < 0.05 by one-way ANOVA.

## Figures and Tables

**Figure 1 molecules-28-02370-f001:**
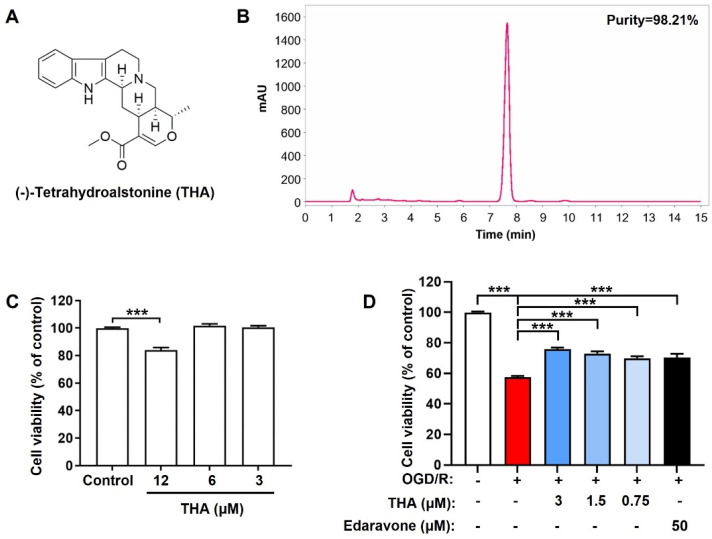
THA protects cortical neurons against OGD/R-induced neuronal injury. (**A**) The chemical structure of THA is shown. (**B**) The purity of THA extracted from *Alstonia scholaris*. (**C**) Cell viability of cortical neurons after THA induction was determined. (**D**) Different concentrations of THA were added in cortical neurons 2 h before and 24 h after OGD/R, and cell viability was determined. Data are expressed as mean ± SEM. *** *p* < 0.001.

**Figure 2 molecules-28-02370-f002:**
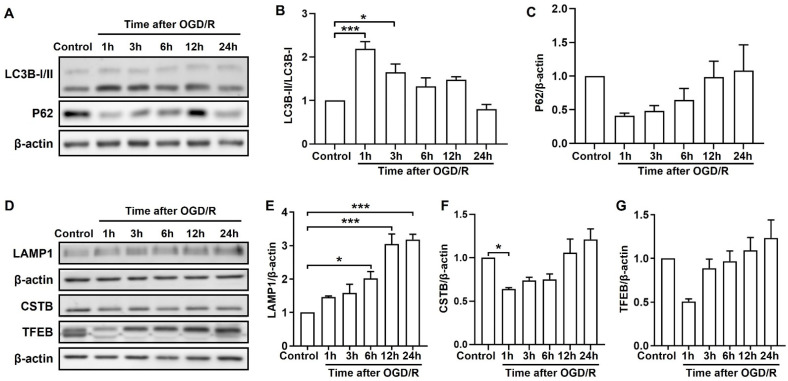
OGD/R induces autophagy and lysosomal dysfunction at different stages. Cortical neurons were subjected to OGD for 4 h followed by re-oxygenation for 1, 3, 6, 12 and 24 h, respectively. (**A**–**C**) Expression and quantification of LC3B-II/I(B) and P62(C) in cortical neurons induced by OGD/R. (**D**–**G**) Expression and quantification of LAMP1 (E), CSTB (F) and TFEB (G) in cortical neurons induced by OGD/R. (**E**–**G**) Expression and quantification of CSTB and TFEB in cortical neurons induced by OGD/R. Data are expressed as mean ± SEM. *** *p* < 0.001, * *p* < 0.05.

**Figure 3 molecules-28-02370-f003:**
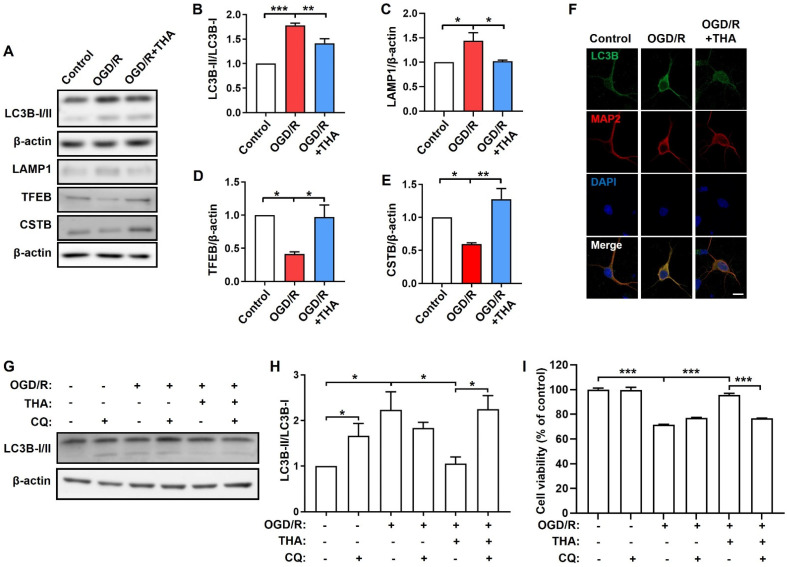
The protective effect of THA against OGD/R-induced neuronal damage occurs via autophagy regulation. (**A**) Cortical neurons were exposed to OGD/R and treated with or without THA (3 µM), and cell lysates were collected for Western blot analysis of LC3B-II/I, LAMP1, CSTB, TFEB. (**B**–**E**) Quantification of LC3B-II/I (**B**), LAMP1 (**C**), CSTB (**D**) and TFEB (**E**) was performed with Image J. (**F**) Immunofluorescence of LC3B and MAP2 (neuronal marker) in OGD/R and treated with or without THA. Scale bar: 10 μm. (**G**–**I**) CQ (chloroquine, 10 µM) blocked the neuroprotection of THA (3 µM) against OGD/R-induced neuronal damage. (**G**,**H**) The expression and relative densities of LC3B-II/I in OGD/R-induced neurons after treated with or without THA and/or CQ. (**I**) Cell viability of OGD/R-induced cortical neurons after treated with or without THA and/or CQ. Data are expressed as mean ± SEM. *** *p* < 0.001, ** *p* < 0.01, * *p* < 0.05.

**Figure 4 molecules-28-02370-f004:**
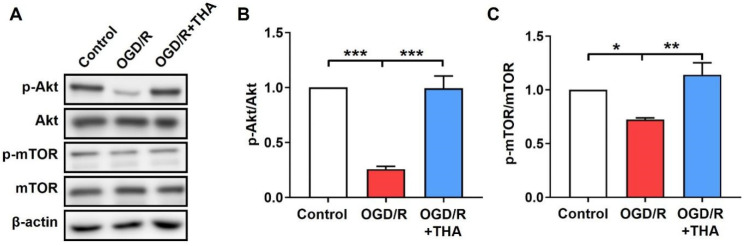
THA restores the activation of Akt/mTOR pathway in OGD/R-induced neurons. (**A**) Cortical neurons were exposed to OGD/R and treated with or without THA (3 µM), and cell lysates were collected for Western blot analysis of p-Akt, Akt, p-mTOR and mTOR. (**B**,**C**) The relative band densities of p-Akt/Akt (**B**) and p-mTOR/mTOR (**C**) were measured. Data are expressed as mean ± SEM. *** *p* < 0.001, ** *p* < 0.01, * *p* < 0.05.

**Figure 5 molecules-28-02370-f005:**
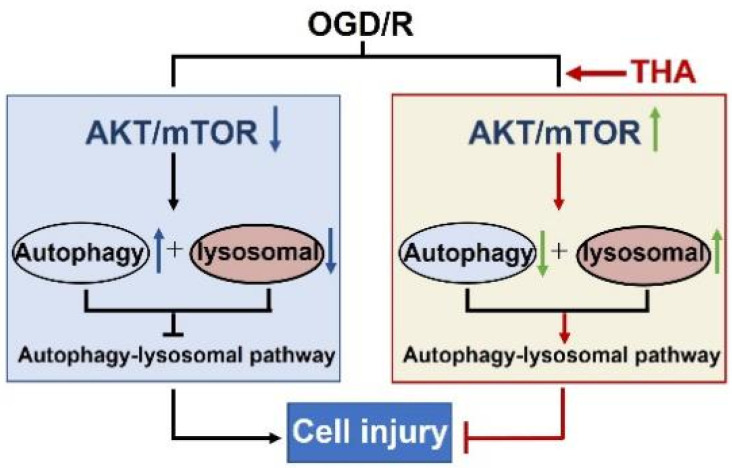
A schematic diagram of THA against OGD/R-induced injury via autophagy regulation.

## Data Availability

The data presented in this study are available on request from the corresponding author.

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
