# Peer review of "The Protective Effect of (-)-Tetrahydroalstonine against OGD/R-Induced Neuronal Injury via Autophagy Regulation"

_molecules, 2023, doi:10.3390/molecules28052370_

Round 1

Reviewer 1 Report

The article is interesting and simple, THA was isolated from Alstonia schorisis and its neuroprotective effect on oxygen-glucose deprivation/reoxygenation (OGD/R)-induced neuronal injury was studied. THA exhibited a favorable protective effect against OGD/R-induced neuronal damage through the regulation of autophagy through the Akt/mTOR pathway.

1. The β-actin in WB is not obvious, and the amount of each sample is also different. β-actin is darker than other proteins.

2. The discussion can go further.

Author Response

The article is interesting and simple, THA was isolated from Alstonia schorisis and its neuroprotective effect on oxygen-glucose deprivation/reoxygenation (OGD/R)-induced neuronal injury was studied. THA exhibited a favorable protective effect against OGD/R-induced neuronal damage through the regulation of autophagy through the Akt/mTOR pathway.

1. The β-actin in WB is not obvious, and the amount of each sample is also different. β-actin is darker than other proteins.

Response: We really appreciate the reviewer to highlight this point. We have replaced the figures of β-actin with a more obvious one in the revised manuscript.

2. The discussion can go further.

Response: We thank the reviewer’s suggestions. We have added some content in the Discussion section in the revised manuscript.

Reviewer 2 Report

In the manuscript entitled “The protective effect of (-)-Tetrahydroalstonine against OGD/R-induced neuronal injury via autophagy regulation” by Liao et al, the authors report the protective effect of Tetrahydroalstonine (THA) extracted from Alstonia scholaris against oxygen–glucose deprivation/re-oxygenation (OGD/R)-induced neuron injury. THA was successfully isolated and characterized with a high purity. Mechanistically, the authors show that autophagy dysfunction is involved in neuron injury induced by OGD/R. Of note, in-vitro studies demonstrate that pretreatment of cortical neurons with THA prevents the neural damage induced by OGD/R, increase cell viability and restore autophagic flux. The topic is quite interesting, especially due to the scarcity of available drugs to treat ischemic stroke. Even though, the manuscript is well written and data are clear, additional experiments would be of great interest to consolidate their conclusion.

1)  Conceptually, the ideal drug to treat ischemic would be one that it is administered after stroke. In the present study, the authors give THA, before inducing neuronal damage. What is the clinical relevance of this strategy? I would suggest the authors to discuss this issue.

2)  Most of the data presented in this manuscript are based on one experimental method, i.e. western blot. I would suggest the authors to use different methods to consolidate their findings. For instance, (i) the increase of the ratio of LC3B-II/LC3B-I can also be confirmed by formation of LC3-positive vesicles (punctate) using LC3 immunolabeling using confocal microscopy; the impairement of autophagic flux can be evaluated by monitoring the expression of proteins that are selectively degraded by autophagy such as sequestosome 1 (SQSTM1); (iii) cell viability can be monitored by alternative method, other than MTT assay;  autophagy activation in OGD/R-induced neuron injury can be monitored by measuring the expression of reduced expression of some autophagy-related (ATG) genes,….

3)  In Figure 2, the authors did a time-course analysis to demonstrate that OGD/R induces autophagy and lysosomal dysfunction. However when addressing the effect of THA in OGD/R, only one time was spotted (Figure 3B-E). To be consistent, I would suggest the authors to do also a time-course analysis, similarly as in Figure 2. In addition, it would be interested to see the cell viability at the same time points.

4)  How do the authors explain, the simultaneous decrease of CTSB and TFEB in their experiments. Normally CTSB is an inhibitor of TFEB (Qi et al, J Exp Med. 2016; 213(10):2081-97. PMID: 27551156). Normally, the downregulation of CTSB is expected to lead to the upregulation of TFEB,

5)  In Figure 3G, THA condition (alone) is missing. In the same line, in Figure 3H, THA condition alone, as well as CQ condition (alone) are missing. It would be great if the authors provide these information

6)  The contribution of Akt/mTOR in THA treatment condition is elusive. Additionally, I would suggest the authors to be precise in the terms used. For example, mTOR. There are two complexes of mTOR, one which inhibits autophagy (mTORC1) and the second which could be involved in the regulation of autophagy (mTORC2). Which one of these complexes the authors are talking about?

7)  Lane 115, the authors claim that “Meanwhile, CQ significantly ameliorated the THA…” (Figure 3 H). This statement is not reflected by the data. Actually, CQ negatively impacts the improvement of cell viability conferred by THA in OGD/R (last histogram of Figure 3H)

Reviewer 3 Report

Protective effect of THA on OGD/R primary cortical neurons of SD rat was evaluated, and THA showed significant protectivity on autophagy and lysosomal dysfunction via Akt/mTOR pathway. I suggest the acceptance of this manuscript upon answering the following question:

1)     Fig.1. THA showed significant cytotoxicity at 12 μM, that means the treatment window was narrow?

2)     Fig.1. why all significance analysis showed “***P<0.001”?

3)     Fig.2. the expression method for significance analysis was different from Fig. 1, Fig 3. and Fig. 4.

4)     A schematic diagram is needed in the Discussion part to show the relationship among  autophagy, lysosomal, Akt/mTOR pathway, etc.

Round 2

Reviewer 2 Report

No additional comments from me. They well and clearly responded to all my questions

Best regards

Author Response

No additional comments from me. They well and clearly responded to all my questions.

Response: We greatly appreciate the reviewer's comments to our manuscript.